# A multi-pathogen behavioral exposure model for young children playing in public spaces in developing communities

Stephanie A. Houser[1]*, Daniel K. Sewell[2], Danielle N. Medgyesi[3], John M. Brogan[4], Jean Philippe Creve-Coeur[5], Kelly K. Baker[3,6]*

**1** Department of Civil and Environmental Engineering, University of Iowa, Iowa City, Iowa, United States of America, **2** Department of Biostatistics, University of Iowa, Iowa City, Iowa, United States of America, **3** Department of Occupational and Environmental Health, University of Iowa, Iowa City, Iowa, United States of America, **4** Terre des Hommes, Lausanne, Switzerland, **5** Terre des Hommes, Port-au-Prince, Haiti, **6** University at Buffalo, State University of New York, New York, New York, United States of America

\* shouser@nas.edu (SAH); kkbaker@buffalo.edu (KKB)

**Data Availability Statement:** The data and code for the model have been made publicly available. The environmental soil data can be downloaded

## Abstract

Diarrheal disease is the second leading cause of death among children under five in developing communities, despite active interventions to improve access to water, sanitation, and hygiene resources. Even interventions with high fidelity and compliance saw minimal improvements in health outcomes, necessitating a need for looking beyond traditionally targeted exposure pathways. This study focuses on the pathogen exposure a young child may receive while playing in the public spaces of informal settlements, where animal feces, human feces, and garbage are frequently observed. The study utilized 79 soil samples previously collected across ten cluster sites in Corail, Haiti and processed using DelAgua cultural assays and quantitative Polymerase Chain Reaction methods. Molecular assays screened for Aeromonas, *Vibrio cholera*, and several pathogenic *Escherichia coli* species. Behavioral observations of young children (<5 years old) were also conducted in overlapping areas with the environmental sampling to quantify frequency of risky and mitigating behaviors. Environmental and behavioral data were combined to obtain the posterior distribution of children's pathogen exposure from playing in a public space for one hour. The model estimated that children have a likelihood of approximately 70% of being exposed to a pathogen during one hour of play and a greater than 30% chance of being exposed to multiple pathogens in the same period. While children and toddlers practice fewer risky behaviors compared to infants, they were shown to have higher likelihood of exposure and slightly higher pathogen dose per exposure. As anticipated, a high correlation between genes from the same *E. coli* species was observed in the model output. These findings demonstrate the need to consider public spaces as an important exposure pathway for young children for both future research and interventions.

from https://myweb.uiowa.edu/dksewell/public/haiti-soil.csv. The behavioral data can be downloaded at https://myweb.uiowa.edu/dksewell/public/haiti-behavioral.csv. The code and code documentation can be found at https://myweb.uiowa.edu/dksewell/pathome/Analysis_for_A_Multi-Pathogen_Exposure_Model_for_Children_in_Developing_Communities.html.

**Funding:** S.A.H. was supported during this work by a grant received by K.B.B. from the National Science Foundation Division of Graduate Education under Grant No. 1633098. This funder had no part in study design, data collection, analysis, publishing, or manuscript development. https://www.nsf.gov/ Some of the laboratory work was funded in part by the PATHOME study, National Institutes of Health Fogarty International Center Grant Number R01TW011795 to K.B.B. at the University of Iowa. This funder had no part in study design, data collection, analysis, publishing, or manuscript development. https://www.nih.gov/ Funds were also received from Terre des Hommes to K.B.B., a local NGO in Haiti for sampling and data collection. There was no specific grant information, and this funder contributed to data collection decisions. https://terredeshommes.org/.

**Competing interests:** The authors have declared that no competing interests exist.

## Author summary

Diarrheal disease is the second leading infectious cause of death among children under five worldwide; however, much is being done through interventions to improve conditions and reduce exposure to pathogens. Despite these efforts, little progress has been made on health outcomes. This study looks at an often-overlooked area in interventions, the public spaces in communities where children frequently play. Soil samples were collected and analyzed for pathogens and children were observed to understand frequency of risky behaviors that could lead to soil ingestion. This data was then fed into a statistical model to replicate thousands of instances of a child playing in a public space for one hour and found that the likelihood of pathogen exposure was high at about 70% with a greater than 30% chance of exposure to multiple pathogens in one instance. This has implications for how we plan interventions to include public spaces. It further serves as a methodologic model for combining site specific behavior and environmental data into a statistical model as a robust way to achieve pathogen exposure frequency and dose.

## Introduction

As efforts continue toward achieving the Sustainable Development Goals (SDGs), the WASH sector needs to be critical of where and how it carries out interventions to achieve Goal 6, aimed at providing all people with access to clean water and sanitation. If done effectively, the containment of human excreta and the protection of water sources will contribute to a decrease in child diarrhea, stunting, and death for children in developing communities, particularly those under five. But recent large interventions of improvements to household WASH have been largely unsuccessful in generating meaningful health impacts despite high levels of fidelity in carrying out the intervention as described and compliance among participants in consistent practice of promoted behaviors or technologies [1].

Several aspects of the enteric pathogen transmission ecosystem in disease endemic settings were not targeted by previous studies. Many of these studies are randomized control trials that focus on household drinking water, creating shared pit latrines, building hand washing stations, nutritional supplements, and/or related behavior changes [1–9]. Despite high fidelity and compliance, meaningful long term impacts were limited to an improvement in linear growth for children born into a clean environment [1–9]. For children transitioned into a more hygienic environment through an intervention, no change in linear growth was seen [1–9]. And finally, improvements were only seen in one study, the WASH benefits trial in Bangladesh, for the short-term outcome of childhood diarrhea [1–9]. Few of these studies looked at public spaces for interventions; therefore, there is a need for interventions and studies that focus on assessing the exposure from and eliminating human and animal feces contamination across the community's public spaces.

Studies have also looked to the exposure pathways often identified in the home including household surfaces, playmats, and drinking water containment; however, these have proven poor indicators of improvement in health outcomes. The WASH Benefits studies found improvements in stored drinking water quality in both sites and reduced food contamination due to handwashing in Bangladesh, but no other effects on any pathways [1]. The Maputo Sanitation project in Mozambique focused on expanding access to improved sanitation infrastructure, but ultimately found that most of the soil and drinking water in the home were still contaminated with fecal indicators and *Escherichia coli* albeit with reduced concentrations,

particularly for instances of improved access to latrines [8,10–12]. The minimal impact of many of these interventions may stem in part from the need for only one interaction by a child with fecal matter to cause infection and minimize the impact of the efforts. This is a problem in the household, but an even greater problem when considering the larger environment that may be played in by children, which is frequently overlooked as an exposure pathway [5,13,14]. Studies have shown that soil contamination is positively associated with prevalence of child diarrhea based on caregiver reported data [15,16].

The narrow focus in many studies on exposure conditions in the household likely contributes to the poor measurements of outcomes as it has been demonstrated that young children frequently spend time playing in public spaces that often contain animal feces, human feces, and garbage. Based on recent observations at 32 public residential sites in a low-income neighborhood of Haiti, young children (<5 years) were present at all sites, including those just beginning to walk (nearly all sites) and infants (approximately half of the sites) [15]. In these public spaces, environments have both a higher concentration and diversity of pathogens than in the home environment [16]. Children playing in these spaces could have been exposed to public latrines (50% of sites), free roaming animals (97% of sites), human feces (36% of sites) and animal feces (100% of sites), trash (100% of sites), and open drainage ditches (56% of sites) [15]. Contact with pathogens present in public spaces not only can put children at high-risk for infection but may also contribute to the contamination of household surfaces, food and water when children return home without handwashing. Although it's plausible that children could be exposed to multiple pathogens in public areas while playing, there are few studies quantifying the likelihood and dose of such exposure [17]. To date, no studies have estimated pathogen-specific exposure during play that accounts for the frequency of empirically measured contact behaviors such as hand-to-object and hand-to-mouth.

Interventions focused on reducing human and animal fecal contamination in public spaces have proven to have impacts on child pathogen exposure or health outcomes. A study in rural Bangladesh that focused on cleaning public latrines, properly disposing of child feces, and increasing access to hand washing stations with soap, demonstrated a reduction in childhood diarrhea prevalence at endline of the intervention [2]. Another intervention that focused on implementing a play-yard in rural Zambian communities to reduce child ingestion of soil and animal feces demonstrated uptake of the technology, but declared a need for further study to identify impact on child outcomes [18]. Many studies confirm the need for better understanding of and reductions in human and animal fecal contamination of public spaces in order to see meaningful health outcomes related to diarrhea and stunting [19–22].

This study aims to quantify the average risk of enteric pathogen exposure and co-exposure that children experience from interacting with soil and objects when playing in public residential spaces in neighborhoods with limited sanitary infrastructure. We hypothesize that child-environment contact in these spaces can increase the risk of exposure to different types of pathogens and that this risk is modified by child motor-development stage. This study combined count data from structured observation of child behavior in July of 2017 and enteric pathogen presence/absence and concentration data from soil collected in a peri-urban neighborhood in Haiti in December of 2016 in a Bayesian simulation to obtain the likelihood of child exposure to multiple pathogens during an hour of play in a public space. The study also aimed to examine how age-based levels of early childhood development (e.g., mobility) influences public play exposure risks. To this end, the aim was to estimate exposure as a function of behaviors displayed across a group of diverse children of different developmental stages, the presence and concentration of pathogens in the environment (including the interactions between multiple co-occurring pathogens), and microbial transference dynamics. While the primary focus was on modeling pathogen-specific exposures, the reliability of general *E. coli*

was also explored, as they are more commonly used as indicators of fecal exposure. In tropical settings, *E. coli* presence may derive from endogenous environment sources as well as human or animal feces, creating significant challenges in interpreting true health risks and fecal sources of ingested *E. coli*. A Bayesian approach allowed for prior knowledge of the system to inform the model, as well as the environmental samples and child observations to provide better inference that includes the uncertainty around this complex system.

## Methods

### Ethics statement

The Institutional Review Board (IRB) at the University of Iowa reviewed and approved the human observation part of this study (ID#201705786) including the researchers not gaining informed consent, as no personally identifying information was recorded, so all data were analyzed anonymously.

### Location description and observed population

Environmental microbiology data was collected in Corail, Haiti, an informal peri-urban settlement in Haiti that developed as a hurricane displacement resettlement 18 miles northeast of Port-au-Prince and now hosts an estimated 14,500 people. The settlement was set up using a common humanitarian camp construction design with rows of community homes with shared latrines and other utilities along the outskirts. There is a seasonal river running along the community that is used for human waste and trash disposal. The site is an ideal place to carry out this study due to the prevalence of feces throughout the entire community due to the free roaming animals, practice of open defecation, and the inability to use most of the latrines due to their pits filling up. In our prior studies we observed substantial heterogeneity in the apparent contamination throughout the site; however, young children were observed playing in nearly all locations [15].

### Sampling collection

To prevent observer bias in site selection and minimize spatial causes of variance in data, sampling sites were selected prior to field data collection. This was performed by randomly selecting eight peripheral sites from a list of twenty sites where community water kiosks and communal latrine blocks were built for sharing by this community at relatively fixed intermittent spatial intervals around the community. A "Cluster" area of approximately 10,000 meters squared was selected near each kiosk that included nearby areas containing latrines and where feces disposal occurs, as well as interior residential areas. In each Cluster area, ten global positioning satellite (gps) coordinates were randomly selected as study "sites" using Batchgeo software (Google). Site gps coordinates were used to orient field teams to the study location, where all area within a fifteen-meter radius around the coordinates was visually inspected and sampled. The combined area of the eighty study sites within the community (~80,000 m$^2$) represents about one fourth of the total community area (~348,000 m$^2$).

The microbial sample collection took place in December of 2016 during the dry season. At each site, the research team used a mobile phone-based structured observational questionnaire, developed using KoboToolbox (Harvard Humanitarian Initiative, Harvard University, Boston, MA), to document the date and time, spatial location, altitude, presence of roads, water kiosks, markets, playgrounds, presence and number of different types of domestic animals, presence and number of adults and children and their activities, percent soil moisture (Exteck Soil Moisture meter, Item # MO750, Nashua, NH, USA), and type of environmental

samples collected. The questionnaire also documented the presence, structural condition, and hygiene of latrines on a site. Observers recorded visual observation of human and animal feces on the ground, as well as evidence of human feces disposal (flying toilets, chamber pot offal, soiled diapers). One soil sample was collected at the center of all sites by using a sterilized metal scoop to collect the top 5 centimeters of soil into a labeled WhirlPak bag. Samples were stored in a cooler on ice pack bags for transport to a make-shift laboratory.

## Sample processing and environmental microbiology

Soil samples were analyzed within 6 hours using DelAgua (Marlborough, Wiltshire, UK) assays according to manufacturer's protocols to quantify *E. coli* bacteria in colony forming units (cfu) per gram of soil. A one gram portion of soil was rinsed for 20 minutes in 10 milli-lites of sterile water, then a 1 mL and 0.1 mL portion were filtered through a 0.45 µM electro-negative gridded Millipore filter and cultured on DelAgua agar. One negative control was run per day alongside the DelAgua tests using molecular grade water. If a positive was observed on these negative controls, retesting would have occurred; however, no positives were observed. A 250 milligram (soil) portion was also measured into a Zymo Shield Lysis tube (Zymo Research, Irvine CA, USA). Samples were subjected to bead beating for 20 minutes to lyse microorganisms and preserve DNA and RNA and then stored at 4˚C until transport at ambient temperature (<24 hours) to the University of Iowa.

Zymo collection tubes with samples were spiked with a $10^5$ concentration of live *E. coli* bacteriophage MS2 (ATCC 15597-B1) as a laboratory extraction control, and then DNA and RNA were extracted using the ZymoBIOMICS DNA and RNA Miniprep kit (Zymo Research, Irvine CA, USA). The sample extracts were then analyzed for target DNA using quantitative Polymerase Chain Reaction (qPCR). The target organisms and respective primer and probe sequences are shown in Table 1.

One microliter of DNA was prepared with 10µL of TaqMan Fast Advanced Master Mix (Thermo Scientific, Waltham, MA, USA), 2µL of primers/probes, and 7µL of nuclease free water in a 96 well plate format for 40 cycles with an annealing temperature of 60˚C and initial hold cycle of 50˚C and then another initial hold cycle of 95˚C. Water only negative controls were run alongside each plate of qPCR reactions. Collection, storage, and extraction efficiency was estimated by subtracting the gene copy concentration of MS2 in samples from the gene copy concentration of MS2 in water-only controls spiked with the same concentration of MS2. Amplification curves and multicomponent plots of all pathogen amplification peaks were

**Table 1. Primer and probe gene sequences for target organisms for quantitative Polymerase Chain Reaction (qPCR).**

| Organism | Gene Target(s) | Forward Primer | Reverse Primer | Probe |
|---|---|---|---|---|
| Aeromonas | Aerolysin | TYCGYTACCAGTGGGACAAG | CCRGCAAACTGGCTCTCG | CAGTTCCAGTCCCACCACTT |
| *Vibrio cholerae* | *hlyA* | ATCGTCAGTTTGGAGCCAGT | TCGATGCGTTAAACACGAAG | ACCGATGCGATTGCCCAA |
| Enteroaggregative *Escherichia coli* (EAEC) | *aaiC* | ATTGTCCTCAGGCATTTCAC | ACGACACCCTGATAAACAA | TAGTGCATACTCATCATTTAAG |
| Enteroaggregative *Escherichia coli* (EAEC) | *aatA* | CTGGCGAAAGACTGTATCAT | TTTTGCTTCATAAGCCGATAGA | TGGTTCTCATCTATTACAGACAGC |
| Enteropathogenic *Escherichia coli* (EPEC) | *eae* | CATTGATCAGGATTTTTCTGGTGATA | CTCATGCGGAAATAGCCGTTA | ATACTGGCGAGACTATTTCAA |
| Enteropathogenic *Escherichia coli* (EPEC) | *bfpA* | TGGTGCTTGCGCTTGCT | CGTTGCGCTCATTACTTCTG | CAGTCTGCGTCTGATTCCAA |
| Enterotoxigenic *Escherichia coli* (ETEC) | *LT* | TTCCCACCGGATCACCAA | CAACCTTGTGGTGCATGATGA | CTTGGAGAGAAGAACCCT |

inspected to verify true amplification. The quantification cycle (Cq) was determined using ViiA7 software.

Pathogen concentrations per milligram of soil were determined by comparing the Cq value of the environmental DNA template to the linear slope of pathogen-specific standard curves. Pathogen-specific standard curves were developed by performing reactions in duplicate on six tenfold serial dilutions of a positive control of known starting concentration (log 10 copies/μl). For *E. coli* species, the known concentration was determined by serial dilution of culture growth in the lab. A gBlock assay of known concentration was used for Aeromonas (ATCC) and *V. cholera* (BEI) controls. The Cq values of averaged qPCR serial dilutions were analyzed by linear regression to establish the slope of the standard curve with a minimum $R^2$ value of 0.90, and to identify the lower limit of quantification (LLOQ) of pathogen gene copies or pathogens that can be reliably detected using qPCR (S1 Fig). Final concentrations per gram were back calculated based upon extraction and dilution. Presence/absence frequencies are reported based upon process limit of detection, while descriptive statistics on concentrations are reported for samples greater than or equal to the LLOQ.

## Behavioral observational data

The quantitative behavioral data used for this study are described in depth in Medgyesi, et al. [15] and were collected in July of 2017 during the wet season (making the data not temporally matched to the environmental data). In brief, at each public site, the behaviors of randomly selected children (n = 386 children) were observed for up to 60 minutes at a time (t = 3,286 total minutes) or until they left the 20-meter radius that defined each site in which case the amount of time of observation was recorded. These sites were observed for four to six hours to capture data on child-environment interactions for a variety of individual children throughout the day. This population-level convenience sampling approach was used to overcome limitations with more traditional enrollment and observation of identified child subjects, including reactivity of caregivers and children to the presence of an observer and potential bias from over- or under-recruitment of children who play in public residential areas. Formal consent was not obtained and no personally identifying information was recorded to preserve child anonymity. The children were categorized by developmental age groups that could influence how behavior and exposure change over development, such as changes in amount of time spent sitting or crawling on the ground touching dirt and spatial range of mobility. Site observations generated behavioral data for 29 infants (not walking independently), 117 toddlers (independently walking but unsteady/not running), and 240 young children (independently walking and running and approximately up to the age of five). During observation, each time the child partook in a specified behavior, the enumerator would mark the start and stop time for that behavior in a tablet-based application. Recorded behaviors included touching soil, surface water, objects on the ground, animal or human feces, eating soil, touching the mouth, drinking surface water, and washing hands.

## Statistical analysis

The primary objective of the statistical analysis was to make inference on the age-stratified dose and pathogen diversity per hour for children. This was obtained using the posterior belief distribution conditional on three data sources: structured observational data, microbiological data, and prior studies from extant literature about microbial transfer rates. Weakly regularizing priors were used for model parameters to help stabilize the estimation and maintain plausible values. We examined this posterior distribution through a sequential sampling Monte

Carlo approach, which can coarsely be represented as the following joint posterior:

$$\pi_4(dose \ / \ hr \ |envir. \ contam., \ behavior, \ transfer \ rates)$$
$$\cdot \pi_3(transfer \ rates \ |external \ data)$$
$$\cdot \pi_2(behavior|struct. \ obs.) \cdot \pi_1(envir. \ contam. \ |micro \ data),$$

where $\pi_1$ corresponds to the posterior distribution for environmental contamination (concentration of a pathogen gene per gram of soil) based on the microbiological data, $\pi_2$ corresponds to the posterior distribution of the behavioral rates given our structured observation data, $\pi_3$ corresponds to the posterior distribution for transference rates based on external studies, and $\pi_4$ corresponds to the deterministic relationship between the dose/hr and behavioral rates, environmental contamination, and transference rates. Our Monte Carlo strategy then involved sampling from $\pi_1$, $\pi_2$, and $\pi_3$ and using these draws obtain a posterior draw of the age-stratified dose of that pathogen per hour from $\pi_4$. This process thus yields samples from the desired marginal posterior distribution, namely

$$\pi(dose \ hr \ |external \ data, \ structured \ observations, \ microbiological \ data).$$

The posterior distribution of environmental contamination ($\pi_1$) was based on a sequential hierarchical modeling approach for the observed microbiological data. Briefly, this approach is based on telescoping out the joint likelihood of all seven pathogen genes for each individual into seven bivariate generalized linear models (GLMs) built sequentially; the $j^{th}$ GLM conditions on the first $j - 1$ pathogen genes already modeled, and thus when taken together represent a joint likelihood for all seven genes. Each bivariate GLM is itself sequential, where first the presence/absence of the pathogen gene is modeled based on a Bernoulli GLM, and then, conditioning on the pathogen gene being present, the $\log_{10}$ concentration is modeled based on a log-normal distribution. The order of the pathogen genes in this sequential modeling approach were set from least to most prevalent in order to obtain the smallest variance of the parameter estimates. Details are provided in the Supplementary Material (S1 Text). This strategy incorporates the relationships between pathogen genes, leveraging the predictive power of the presence/absence/concentration of one gene on the presence/absence/concentration of the others. That is, our modeling strategy predicts combinations of pathogens present in a sample based upon gene indicators, in contrast to predicting the presence of one pathogen at a time.

The posterior distribution of behaviors ($\pi_2$) was obtained in the following way. For each behavior, we fit hierarchical generalized linear models (HGLMs) based on the following four distributions: Poisson, negative binomial, zero inflated Poisson, and zero inflated negative binomial [15]. Each HGLM used age group as a categorical covariate (infant, toddler, child), and placed a common normal prior on the corresponding regression coefficients; the log of the time each child was observed was fixed as an offset term. We then used Bayes factors to choose the most likely model that describes each of the behaviors. We then sampled from the predictive posterior distributions for the behavioral counts in one hour.

Transference parameters were drawn according to distributions given in the literature($\pi_3$). These selected distributions are listed in Table 2. Given a child's behaviors, the environmental microbial contamination, and the transference parameters, a child's exposure during an hour

**Table 2. Variables, parameters, and values used for the dose calculations.**

| Data | | Variable | Type | Values |
|---|---|---|---|---|
| Transfer efficiency from soil to hand [23] | | $t_{e,s\text{-}h}$ | Log normal distribution | $\mu = 0.11$, $\sigma = 2.0$ |
| Transfer efficiency from object to hand [24] | | $t_{e,o\text{-}h}$ | Normal distributed parameter | $\mu = 0.23$, $\sigma = 0.22$ |
| Transfer efficiency from hand to mouth [25] | | $t_{e,h\text{-}m}$ | Beta distribution | $\alpha = 5.2$, $\beta = 2.6$ |
| Transfer efficiency from object to mouth [23] | | $t_{e,o\text{-}m}$ | Beta distribution | $\alpha = 2$, $\beta = 8$ |
| Child hand surface area [26] | $cm^2$ | $SA_c$ | Triangular distribution | min = 139.63, max = 153.28, mode = 146.47 |
| Toddler hand surface area [26] | $cm^2$ | $SA_t$ | Triangular distribution | min = 80.01, max = 177.38, mode = 121.18 |
| Infant hand surface area [26] | $cm^2$ | $SA_i$ | Triangular distribution | min = 74.59, max = 119.18, mode = 98.31 |
| Object surface area [26] | $cm^2$ | $SA_o$ | Exponential distribution | $\lambda = 0.11$ |
| Concentration of pathogen | Cells/ g | $y_{i,j}$ | Sequential bivariate GLMs | This study |
| Behavior: hand to mouth touch [15] | $hr^{-1}$ | $B_{h\text{-}m}$ | Negative binomial | This study and Ref. 15 |
| Behavior: hand to object touch [15] | $hr^{-1}$ | $B_{o\text{-}h}$ | Negative binomial | This study and Ref. 15 |
| Behavior: geophagy [15] | $hr^{-1}$ | $B_{geo}$ | Zero-inflated Poisson simulated | This study and Ref. 15 |
| Behavior: hand to soil touch [15] | $hr^{-1}$ | $B_{s\text{-}h}$ | Negative binomial | This study and Ref. 15 |
| Behavior: object to mouth [15] | $hr^{-1}$ | $B_{o\text{-}m}$ | Negative binomial | This study and Ref. 15 |
| Handful of soil ingested [26] | g | Hf | Beta distribution | $\alpha = 5.3$, $\beta = 158.6$ |
| Hand loading [26] | $g/cm^2$ | $L_h$ | Triangular distribution | min = 0, max = 0.00095, mode = 0.0001 |
| Object loading [23] | $g/cm^2$ | $L_O$ | Log normal distribution | $\mu = 0.11 + \log(0.001)$, $\sigma = 2$ |
| Fraction of hand touching soil [27] | | $F_{s\text{-}h}$ | Normal distribution | $\mu = 0.230$, $\sigma = 0.108$ |
| Fraction of hand touching mouth [27] | | $F_{h\text{-}m}$ | Normal distribution | $\mu = 0.193$, $\sigma = 0.073$ |
| Fraction of hand touching object (assumed similar to hand touching soil) | | $F_{o\text{-}h}$ | Normal distribution | $\mu = 0.160$, $\sigma = 0.025$ |
| Maximal object loading [23] | $g/cm^2$ | $L_{O,MAX}$ | Uniform | min = 6 max = 8 |

of play was a deterministic function ($\pi_4$) given by the equation:

$$\frac{Dose}{hr} = \left(y_{i,j}\right)\big[\left(t_{e,s-h}\right)(SA_c)(F_{s-h})(F_{h-m})\left(t_{e,h-m}\right)(L_h)(\min(B_{s-h}, B_{h-m}))$$
$$+\left(t_{e,o-h}\right)(SA_c)(F_{o-h})(F_{h-m})\left(t_{e,h-m}\right)(L_h)(\min(B_{o-h}, B_{h-m}))$$
$$+\left(t_{e,o-m}\right)(SA_o)\left(\min\left(L_o, L_{O,MAX}\right)\right)(B_{o-m}) + (Hf)\left(B_{geo}\right)\big],$$

where the variables, parameters, and values used can be found in Table 2. This calculation assumes a positive direction of time-based actions by taking the minimum value of dependent touches. This assumption is used due to the extreme lack of knowledge about how transfer efficiencies respond to multiple touches and the complexity of understanding time relationships.

## Results

### Soil contamination

High quality DNA was available for 79 of 80 collected soil samples. Raw data from both the qPCR and the *E. coli* culture quantification is shown in Table 3. All qPCR negative controls were negative for all of the pathogens tested. Pathogenic *E. coli* genes were detected more often, ranging from 2–38 of the samples, when compared to Aeromonas genes at five samples

**Table 3. Pathogen gene and *E. coli* colony detection frequencies and concentrations per gram of soil from residential areas of Corail.** Means, medians, and range estimated from positive samples and rounded to nearest whole number.

| | *Aeromonas* | *Vibrio Cholera* | EPEC *Eae* | EPEC *bfpa* | EAEC *aaiC* | EAEC *aatA* | ETEC *LT* | *E. coli* culture |
|---|---|---|---|---|---|---|---|---|
| Detection (of 79 samples) | 5 | 9 | 38 | 14 | 13 | 2 | 22 | 59 |
| Mean concentration (pathogens/g) | 5,478 | 484,112 | 306,567 | 10,996 | 478,779 | 206,474 | 18,851 | 4429 |
| Median concentration (pathogens/g) | 130 | 477,807 | 3,928 | 2,368 | 934 | 206,474 | 644 | 1,920 |
| Range of concentrations (pathogens/g) | 85–19,938 | 111,195–873,786 | 689–6,947,445 | 210–75,628 | 455–6,200,779 | 119,515–293,432 | 79–385,419 | 10–35,500 |

or *V. cholerae* genes at nine samples. However, *E. coli* genes were detected at much lower rates than *E. coli* bacteria colonies, which were detected in 59 of the 79 samples.

Comparison of each sample's culture colony counts and the qPCR concentrations demonstrated that qPCR concentrations were often lower than *E. coli* cfu concentrations and were more likely to be non-detects, which may be due to their higher volume-adjusted assay limits of detection, which can be found in the Supplemental Material (Fig 1, S1 Table). The 1:1 line shown transecting the plot space would represent perfect agreement between a single pathogen gene and the culture count.

Cholera toxin gene copies in soil were typically higher than *E. coli* cfu counts as well as gene counts with a median of 477,807 *hlyA* gene copies/g compared to 644 to 206,474 copies/g for pathogenic *E. coli* genes. While a small percentage of the samples had a higher EAEC (23%) and EPEC (26%) gene copy concentration, most samples had a higher *E. coli* culture colony count at an average of 4,429 colonies per gram of soil than pathogen gene concentration. And for all samples, this *E.coli* culture colony count was higher than the gene copy concentration for ETEC *LT*.

To understand the relationships between pathogens and to provide credibility to our qPCR results, we analyzed the correlations between the qPCR gene targets, *E. coli* cfu, and the number of pathogens found through qPCR using the Pearson rank correlation coefficient (Fig 2). There were no notable inverse correlation relationships. There was a strong correlation (>0.7) between the two EPEC genes at 0.78 and each with the EAEC chromosomal gene, *aaiC*, at 0.81 and 0.92, respectively. There was a moderate correlation (0.3 to 0.69) between several of the *E. coli* genes and the *E. coli* culture and between the *Cholera* gene, both EPEC genes, and both EAEC genes and the number of pathogens present in a soil sample.

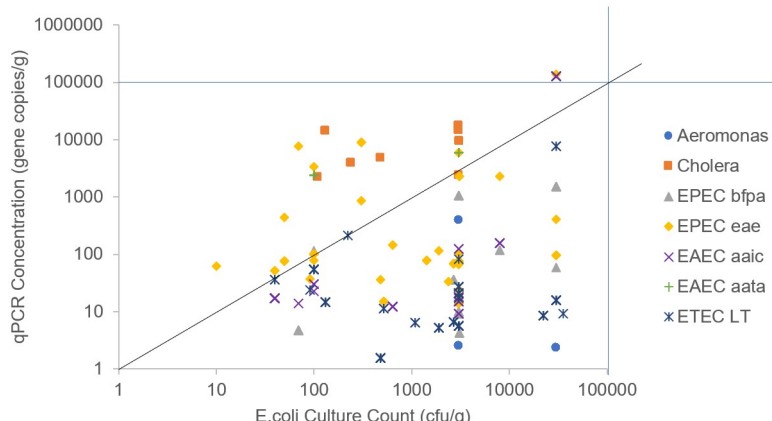

**Fig 1. qPCR concentrations of each pathogen gene versus the *E. coli* culture counts in one gram of soil.**

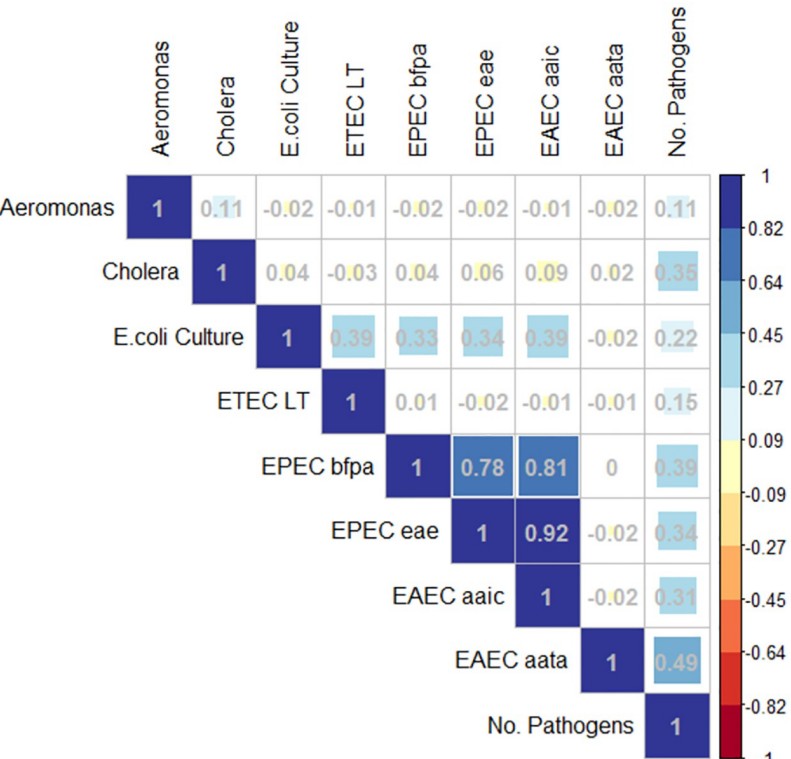

**Fig 2. The Pearson's rank correlation coefficients for comparison of qPCR pathogen gene copies/g and *E. coli* cfu/g.** The number of pathogens variable is an integer representing how many gene targets were present for a given sample based on the qPCR results.

## Pathogen Exposure Assessment

Data used from Medgyesi et al. [15] are summarized in Fig 3 as density functions of observed behaviors according to the child's developmental age group and the length of time observed for each child. Summary statistics are provided in Supplemental Materials (S1 Text). Fig 3 illustrates that all behaviors are zero inflated across all ages, and that there is a dropoff in frequency of behaviors for children compared to infants with toddlers falling in between the two categories. While most observations were less than 10 minutes across all three age categories, there were a substantial number of more prolonged observations (41% for infants, 36% for toddlers, and 23% for children).

Pearson's rank correlation coefficients were calculated for all behaviors for each age category compared to the time of observation (Table 4). As one would expect, positive correlations indicating increased chances of measuring behavior over time were seen for all behaviors across all age categories (except toddler geophagy due to extreme lack of data). Most correlation coefficients were moderate (0.377–0.610) with hand to object touches having a more pronounced correlation (0.726–0.779). Geophagy and mouth to object displayed weaker correlation due to lower frequency of occurrence (all < 0.431).

To understand the risk of pathogen exposure (general *E. coli* assessed by culture excluded) to young children in a peri-urban Haitian neighborhood, we analyzed exposure dose of each of the target organisms individually and together as a multi-pathogen risk incorporating child behaviors. The EPEC *eae* gene were the most common pathogen exposure across developmental age groups (posterior probabilities of 0.481 for infants; 0.474 for toddlers; 0.468 for

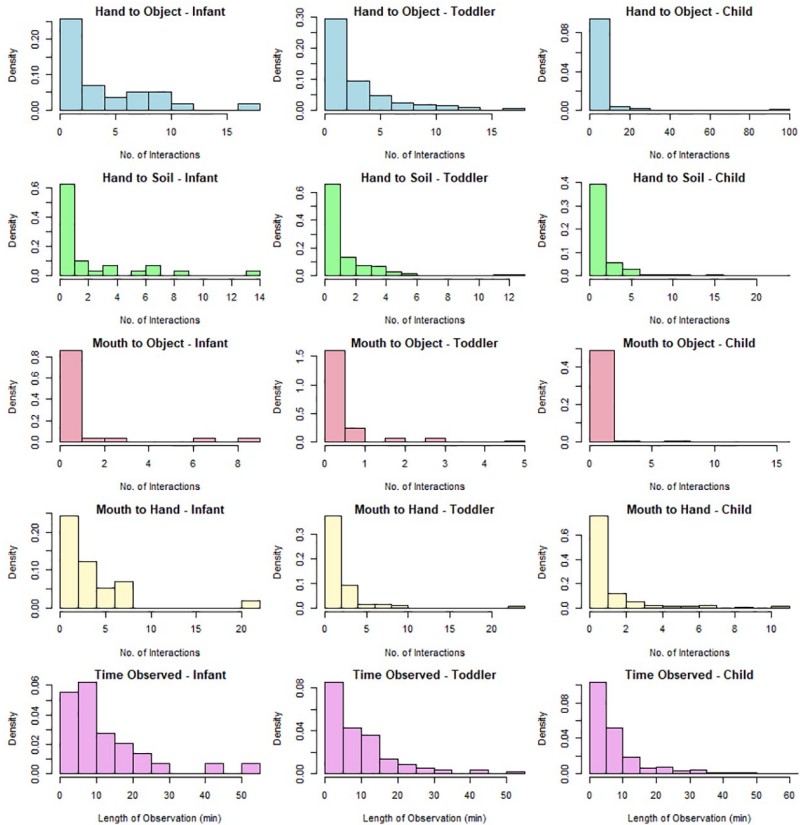

**Fig 3. Frequencies of child exposure behaviors.** The top four rows are histograms of the density function for the frequency of interactions for each behavior per observation session stratified by developmental age category. The fifth row of histograms shows the density function for the length of time observed during an observation session of a child stratified by developmental age category. Geophagy was left out of the behaviors illustrated here due to the extremely low frequency of occurrence, especially for toddlers and children.

children) and EAEC *aatA* was the least common (posterior probabilities of 0.0301 for infants; 0.0295 for toddlers; 0.0289 for children) (Fig 4). All other gene target likelihoods fell between these two with ETEC *LT* being the next highest and *Aeromonas* being the next lowest likelihood of exposure. Additional statistical values can be found in the Supplemental Materials (S1 Text).

For each of the pathogens there was a high posterior probability of a zero dose of a given pathogen over one cumulative hour. Despite having lower presence likelihoods, *Cholera* and EAEC *aatA* had the highest average exposure doses across all age groups (Fig 5). *Aeromonas*

**Table 4. Pearson's correlation coefficients for each behavior frequency compared to the time of observation for a given child's observed play session in the public space.**

|  | **Infant** | **Toddler** | **Child** |
|---|---|---|---|
| **Hand to Object** | 0.7787 | 0.72607 | 0.75893 |
| **Hand to Soil** | 0.5668 | 0.37715 | 0.54353 |
| **Mouth to Object** | 0.2147 | 0.20806 | 0.43058 |
| **Geophagy** | 0.1417 | -0.02790 | 0.16796 |
| **Mouth to Hand** | 0.4376 | 0.44068 | 0.61042 |

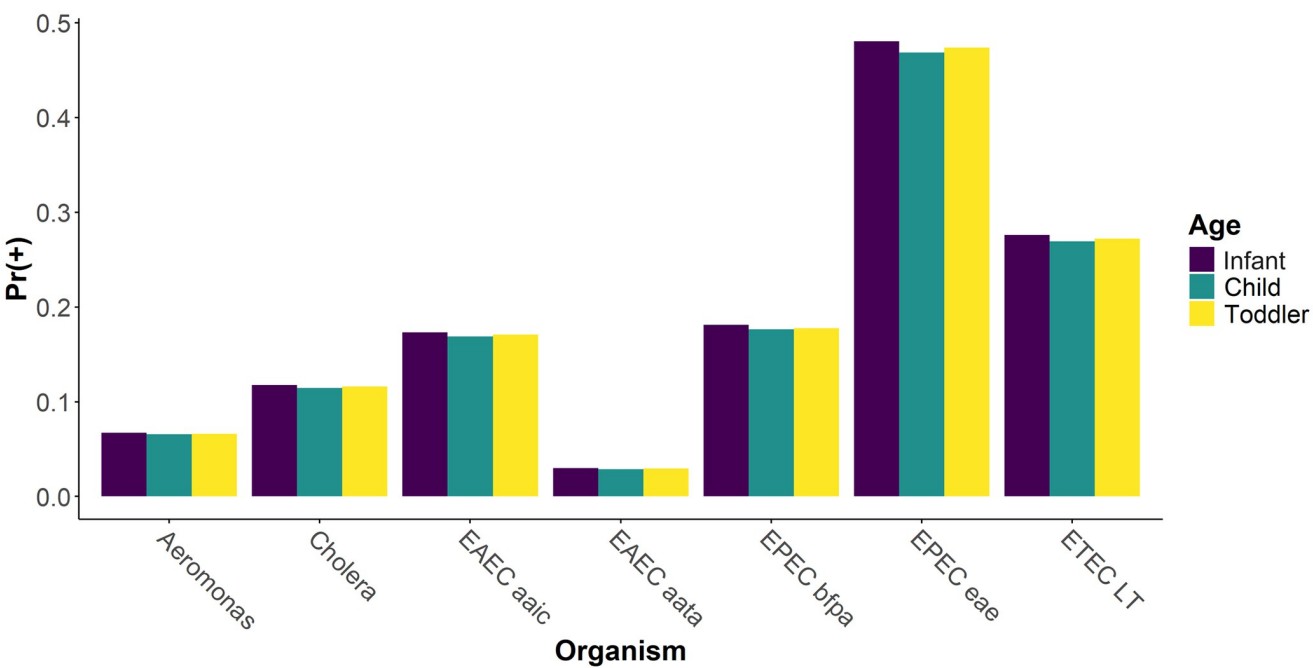

**Fig 4. Probability of exposure of each pathogen gene by developmental age group.** Additional statistical information (median, 95% confidence interval, etc.) can be found in the Supplemental Materials (S1 Text).

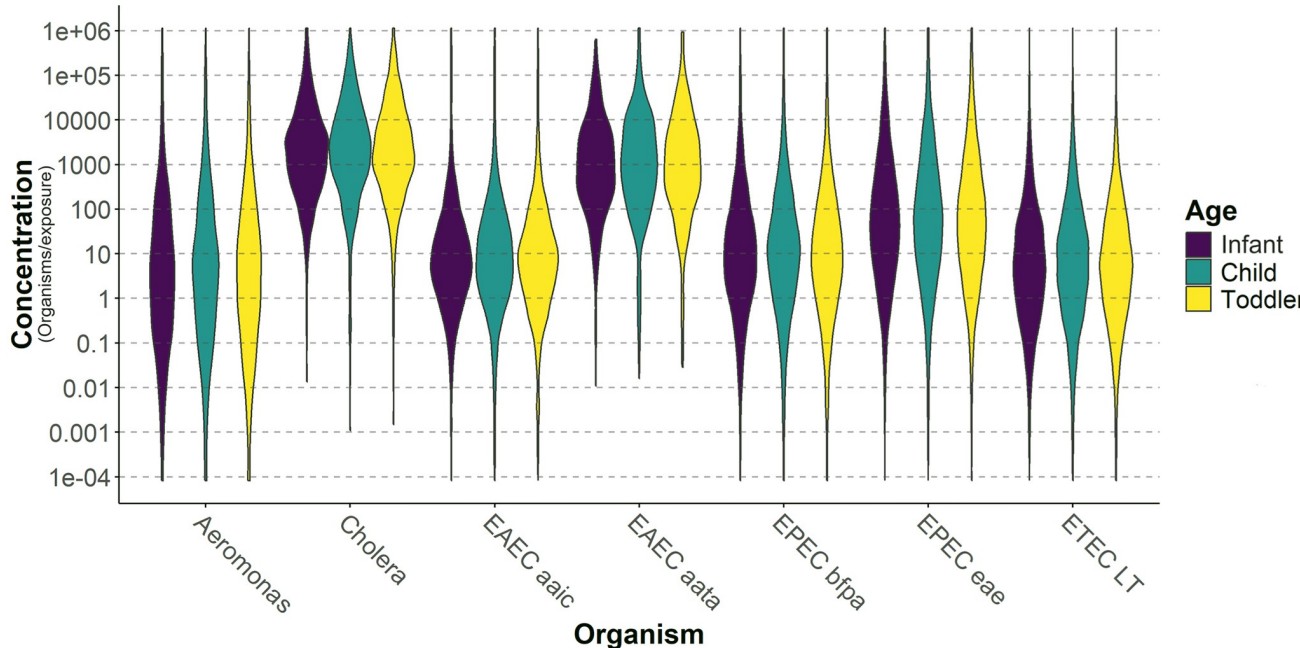

**Fig 5. The exposure dose stratified by developmental age category when a pathogen was present.** Each violin plotted represents the density curve for a given dose and age category with the wider parts of the violin corresponding to the greater frequency of pathogen doses at that concentration.

consistently had the lowest average dose of organisms due to a wider spread in its exposure dose. While difficult to see in graphical representation, there are slight increases in the probability and dose of pathogen exposure as age category increases (average infant dose < average toddler dose < average child dose). The two older and more mobile groups of children experienced a higher likelihood of pathogen exposure with an increase in average concentration across all pathogens, despite participating in the lowest frequency of risky behaviors, like mouthing hands and mouthing objects, of all three developmental categories [15].

Beyond analysis of individual pathogens, we analyzed the probability of multiple pathogen exposure, combining *bfpA* and *eae* genes as EPEC positive and *aatA* and *aaiC* genes as EAEC positive, being ingested during the same hour of play, which could cause comorbidity and increase the risk of disease [28,29]. During one hour of play in a residential public area, all developmental age categories had an approximately70% chance of ingesting at least one pathogen; however, they further had a greater than 30% chance of ingesting two or more types of pathogens at a time (Fig 6). Differences between developmental age groups were small, but showed a slight decrease in likelihood of multi-pathogen exposure as children reached older age stratifications.

The multivariate model elucidates risk probability by accounting for correlative relationships between the different pathogens. The model demonstrated a moderate correlation between exposure to the EPEC *eae* gene target and the EAEC *aaiC* gene target at 0.36, between exposures to *Aeromonas* and *Cholera* gene targets at 0.32, between exposures to the EAEC *aaiC* gene target and the EPEC b*fpA* gene target at 0.54, and between exposures to the two EPEC gene targets at 0.53 (Fig 7). There was also a weak correlation between exposure to the several of the *E. coli* target genes and the *Cholera* target gene, ranging from 0.14–0.19. Unexpectedly, no correlation was observed between the chromosomal EAEC gene target (*aaiC*) and the EAEC plasmid gene target (EAEC *aatA*).

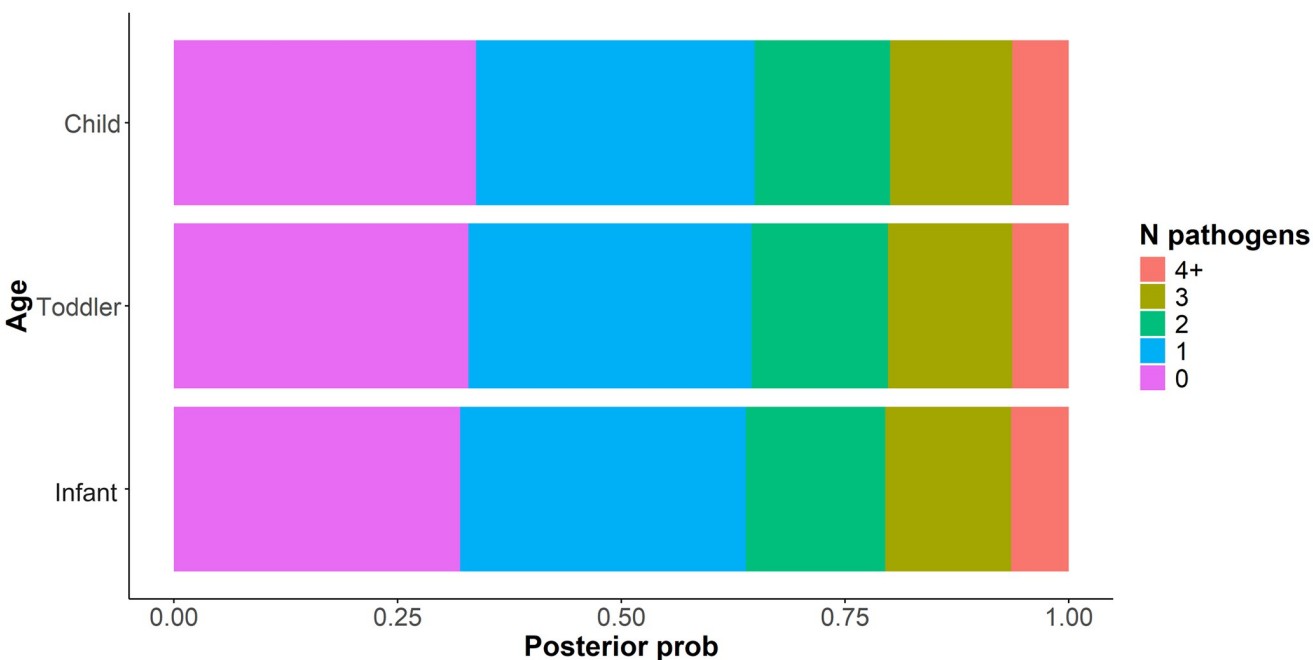

**Fig 6. The probability of exposure to genes indicating different pathogen types, by developmental age group.**

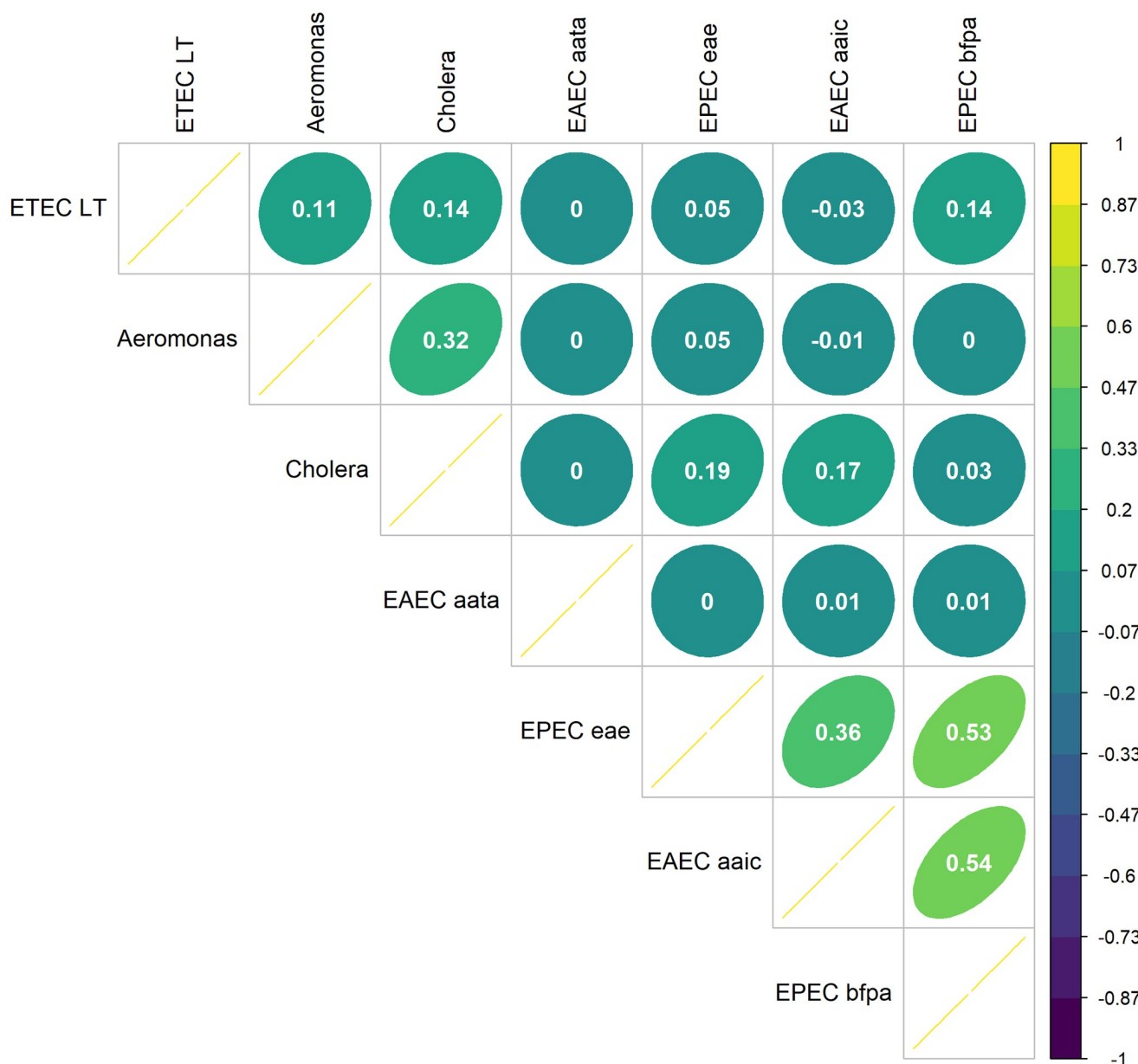

**Fig 7. The Spearman's correlation coefficient between exposure doses for the target pathogen genes modeled in this study.**

## Discussion

This study demonstrates the relatively high likelihood of pathogen exposure for children playing in this peri-urban settlement, especially when considering that most children play for more than one hour a day and exposures presented here only model one hour. While it is difficult to draw conclusions on disease burden due to doses modeled in this study as a result of the lack of existing data and the wide variability of dose response for enteric pathogens, these doses are at the perceived infectious level for adults, making it even more likely for infection of a child [30]. Relative to typical standards for child consumption of safe drinking water which requires no detection of *E. coli* bacteria, these exposure rates and doses very high. Further,

some differences by developmental age stratification emerge when comparing both likelihood of a single pathogen exposure, likelihood of multiple pathogen exposure, and concentration of a given pathogen that is ingested. Several reasons could explain these differences, including increased hand size as children grow, increased hand-mouth motor coordination, increased curiosity in their environment, and increased mobility. Other studies support these developmental factors and, in agreement with our results, have found that mouthing behaviors are more common in the first two years of life while soil touching behaviors increase with age [31–33]. More work is needed to further understand exposure likelihood and differences by age, but most importantly, the universality of these findings in other settings, due to the extreme heterogeneity of microbial contaminants in these peri-urban informal settlements.

This paper demonstrates a methodology that is critical for understanding microbially-complex enteric pathogen exposure risks to children from playing in public spaces where there is a lack of safely managed sanitation service in the community. However, this approach could be expanded to examine other microbial or chemical health risk analyses. When analyzing multiple pathogens or other contaminants with both presence/absence data and concentration data, a multivariate model such as the one presented here allows for the interactions between pathogens as well as interactions between the presence/absence patterns and pathogen concentrations. It could further allow for multiple outcomes to be identified and analyzed in one model. The zero inflated models used for the behavioral data allow for the increased percentage of absence data to be accounted for within other models (i.e., Poisson, negative binomial). By utilizing Bayesian inferential methodology, we can seamlessly integrate knowledge from prior external studies, and can provide interpretable, quantifiable results. This approach could also be extended to examine other exposure risks in other settings, for example oil clean-up crew exposure after an oil spill like Deep Water Horizon or for clean-up crews at Superfund sites like the exposure to amphibole asbestos in Libby, MT, USA [34,35].

One of the statistical limitations of this study was the high level of environmental contamination present in the neighborhoods. The co-occurrence of multiple types of animals and animal feces in nearly all locations and the presence of human defecation and latrines that do not safely separate excreta from the environment makes discerning strong relationships or drawing conclusions about the causation of the contamination challenging without larger sample sizes and study designs supporting causal inference. These heavily contaminated spaces could also influence the likelihood of child behaviors of interest in this study, for example the presence of trash could motivate a child to play in a given space or practice certain behaviors. Heavy trash contamination could also incentivize a caregiver to not let a child play in that space. The observations of this study were not designed to shed light on this feedback process and further research would be needed.

Another limitation is that this study only utilized five pathogens with seven gene targets, which decreases the potency of the multi-pathogen exposure results. There are hundreds of enteric pathogens that can be spread via human and animal feces. Due to the limited number of pathogens modeled, there is a reduced number of relationships that can be found and there is less chance for discovering multi-pathogen exposure. This study demonstrates the viability of such a model to analyze this complex problem that is often associated with contamination data having both presence/absence, concentration, and the potential for symbiotic or competitive relationships to exist between pathogens. Screening for and modeling other or more pathogen genes may have yielded stronger correlative relationships. Some of the low correlation rankings between culture and qPCR methods for detecting *E. coli* could be due to lack of screening for other types of pathogenic *E. coli* like Shiga-toxin expressing (STEC/EHEC), diffuse adherent (DAEC), or enteroinvasive (EIEC) *E. coli*. Most studies have reported detection of STEC or EHEC in soils is uncommon, but EIEC/*Shigella* spp. has been detected in similar

contaminated communities in Kenya and Mozambique. In these studies that screened for more diverse pathogen gene targets, low correlation with *E. coli* cfu was also noted, supporting the premise that *E. coli* alone is not enough to be the global reference indicator for fecal contamination [8,11,17,36].

The sampling and microbial methodologies used in this work also come with limitations in regards to their predictive power of true concentration. There are a wide range of *E. coli* species that would appear on a culture yet not be pathogenic, including human and animal gut microbiome and environmental strains [37]. General *E. coli* therefore may have low predictive power for pathogenic *E. coli* presence. In contrast, the qPCR methodology demonstrates pathogenicity of a bacteria by searching for gene targets only present in pathogenic strains, but does not ensure viability of a pathogen, as it is possible to pick up dead or extracellular DNA with qPCR. Both of these could result in an overestimation of the live pathogenic organism exposure dose of a child. The discrepancy between culture and qPCR concentrations could also be attributed to a combination of *E. coli* species subtypes common to human, animal, and environmental microbiomes all contributing to the colony count but only one species contributing to the gene copy concentration.

The choice to model only one hour of time a child spends playing outside limits the accuracy of daily exposure estimates. While many children, especially of mobile age, in these settings spend parts of their day in public spaces, evidence on cumulative time per day experiencing public exposure is lacking [15]. Their exposure may be much greater than what is expressed in this study. The choice to limit exposure time was made to allow for modeling of exposure from a single soil sample, as children likely move between different areas of the public space, but also to avoid overestimating exposure. Modeling heterogeneity in public exposures would require more evidence on daily child movement in the environment and the modeling of multiple different soil samples within a given exposure dose. Our choice of study design also meant that the total observed time for any random child varied based upon how long a child remained at the observation location. We may not have observed rare behaviors, like geophagy, among children who practice these behaviors but were present for short periods of time. This may have biased exposure estimates for those rare behaviors.

Further, this study was limited by the collection of environmental data and behavioral data being collected seven months apart in different seasons. While it is hard to know the full extent of impact, one might assume that behaviors would be higher in dryer seasons when children are able to play outside more and that environmental concentrations would be higher during wetter seasons when floods and rains can easily transport pathogens. As such both datasets are conservative in their respective contributions to exposure measurements; however, this would be an area worth exploring in future work on this topic.

The work is also limited by the inability to draw health conclusions from exposure calculations. Quantitative Microbial Risk Assessment (QMRA) methodology, which attempts to estimate the probability that an exposure causes health outcomes, incorporates a dose response curve that indicates what dose of pathogen is necessary to cause a certain disease outcome. There are not appropriate dose response curves for the pathogens used in this study and age and geography of the observed population. Dose response curves are typically established from data based in adult populations with very low disease burden. These curves are likely not a good approximation of the actual dose necessary to cause disease in disease endemic populations or in children. In high contamination environments, the human immune system may respond to increased exposure to pathogens by developing higher resistance or a more robust immunoprotective or immunoregulatory response, meaning the exposed individuals would need a higher dose of a given pathogen to result in disease or may not develop disease at all

[38]. Studies of serological responses to enteric pathogens in similar high-contamination environments suggest children develop antibodies to these pathogens early in life, although the effectiveness of those responses in infection prevention is unclear. It is also possible that a high contamination environment could lead to chronic infection, ultimately blunting the individual's immune response and making disease more likely. Further, geographically specific factors like the local diet, local weather, and previous disease prevalence in communities could also contribute to differences in dose response curves [39]. Despite being unable to bring our study to health outcome endpoints due to the lack of relevant dose response curves, it is largely understood that exposure, especially to multiple pathogens, for children under five can be a major health concern at doses shown in this study [40–43].

To continue building on this work, models such as this need to be utilized across various settings and with more pathogens to build up the robustness of the model and to help better understand the relationships between pathogens in different settings. It would also improve the broader understanding of the results to create localized dose response curves to allow for conclusions to be drawn about health outcomes. Dose response curves focused on child response would also be informative for future studies, yet there remain ethical concerns around development of such curves.

## Supporting information

**S1 Fig. Standard curves for qPCR.**
(TIF)

**S1 Table. Lower Limits of Quantification for Environmental Sample Processing using qPCR.**
(XLSX)

**S1 Text. Statistical model code, explanation, and additional results.**
(PDF)

## Acknowledgments

The authors would like to acknowledge Reid Senesac for assistance with environmental sample collection and processing. Additional gratitude to the Haiti team, including Benito Baptiste, Belforde Boully, Marie Carmelle Bruny, and Ketia Rémy, for their local knowledge and collection of behavioral data.

## Author Contributions

**Conceptualization:** Stephanie A. Houser, Danielle N. Medgyesi, John M. Brogan, Jean Philippe Creve-Coeur, Kelly K. Baker.

**Data curation:** Stephanie A. Houser, Danielle N. Medgyesi, John M. Brogan, Jean Philippe Creve-Coeur.

**Formal analysis:** Stephanie A. Houser.

**Funding acquisition:** Kelly K. Baker.

**Investigation:** Stephanie A. Houser, Danielle N. Medgyesi, John M. Brogan, Jean Philippe Creve-Coeur.

**Methodology:** Stephanie A. Houser, Daniel K. Sewell, Kelly K. Baker.

**Resources:** Kelly K. Baker.

**Software:** Stephanie A. Houser, Daniel K. Sewell.

**Supervision:** Kelly K. Baker.

**Validation:** Stephanie A. Houser, Daniel K. Sewell.

**Visualization:** Stephanie A. Houser.

**Writing – original draft:** Stephanie A. Houser, Daniel K. Sewell.

**Writing – review & editing:** Stephanie A. Houser, Daniel K. Sewell, Kelly K. Baker.

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
