## [Decision Letter · Decision Letter 0]

24 Apr 2024

Dear Dr. Houser,

Thank you very much for submitting your manuscript "A Multi-Pathogen Behavioral Exposure Model for Young Children Playing in Public Spaces in Developing Communities" for consideration at PLOS Neglected Tropical Diseases. As with all papers reviewed by the journal, your manuscript was reviewed by members of the editorial board and by several independent reviewers. In light of the reviews (below this email), we would like to invite the resubmission of a significantly-revised version that takes into account the reviewers' comments. 

I believe the comments of reviewer one can be addressed with major revisions, while those of the other two will require minor revisions.

We cannot make any decision about publication until we have seen the revised manuscript and your response to the reviewers' comments. Your revised manuscript is also likely to be sent to reviewers for further evaluation.

Sincerely,

Josh M Colston, Ph.D.

Academic Editor

Justin Remais

Section Editor

I believe the comments of reviewer one can be addressed with major revisions, while those of the other two will require minor revisions.

Reviewer's Responses to Questions

**Key Review Criteria Required for Acceptance?**

**Methods**

-Are the objectives of the study clearly articulated with a clear testable hypothesis stated?

-Is the study design appropriate to address the stated objectives?

-Is the population clearly described and appropriate for the hypothesis being tested?

-Is the sample size sufficient to ensure adequate power to address the hypothesis being tested?

-Were correct statistical analysis used to support conclusions?

-Are there concerns about ethical or regulatory requirements being met?

Reviewer #1: Motivation: The manuscript motivated the importance of identifying and quantifying exposure. The authors relate this to lack of widespread effectiveness of WASH interventions in various studies. However, there are several recent studies that do find associations between soil ingestion and child diarrhea. 

https://adc.bmj.com/content/104/Suppl_2/A116.1

https://www.ncbi.nlm.nih.gov/pmc/articles/PMC5361529/

https://onlinelibrary.wiley.com/doi/full/10.1111/tmi.13510

Given these studies it is not entirely clearly the novelty of the presented manuscript. Further, as the authors highlight the inability of the study to draw health conclusions from exposure calculations – it is not clear how the study can help improve WASH interventions.

Reviewer #2: Clarity is needed on the negative controls run for cultured E coli by DelAgua; the model inputs particularly for the parameters associated with contact frequencies (hand to mouth, object to mouth, hand to surface, hand to soil etc.,) need to be numerically included in the table 1; clearly describing that these are not matched soil and observation data is critical. Additionally more detail is needed on the observation time, extrapolation to 1 hour, and the assumptions associated with this.

Reviewer #3: -Are the objectives of the study clearly articulated with a clear testable hypothesis stated?

The objectives of the study are clearly articulated, but the hypothesis is not explicitly stated. Though not necessarily a numerical estimate, a written hypothesis regarding the risk of exposure utilizing the presented model may help clarify the purpose and context of the study.

-Is the study design appropriate to address the stated objectives?

The study design is appropriate. This multidisciplinary approach obtains the data necessary to create a model that is a function of environmental availability of pathogens, child behavior, and microbial transfer rates. Though the soil samples were collected during the dry season 7 months prior to the time of child behavior observations during the wet season, this limitation and its potential implications are clearly explored in the Discussion section.

-Is the population clearly described and appropriate for the hypothesis being tested?

The population is described in the Results rather than Methods. Though a prior study is referenced, the authors may consider adding a summary of participant enrollment in the Methods section.

-Is the sample size sufficient to ensure adequate power to address the hypothesis being tested?

No sample size or power calculations are included. The authors may share any pre-analysis power calculations or add post-hoc calculations to the supplemental files. Please note that post-hoc power calculations using observed effect size estimates may be noisy (Gelman 2019, DOI 10.1097/SLA.0000000000003089).

-Were correct statistical analysis used to support conclusions?

 A Bayesian multivariate model seemed appropriately employed to support the main conclusions. A Pearson’s rank correlation coefficients were presented to confirm credibility of qPCR results as well. Further detailed review of the manuscript by a biostatistician is needed.

-Are there concerns about ethical or regulatory requirements being met?

There are no significant concerns, but no clear description of the Institutional Review Board approval and consent process are included in this manuscript. These points are outlined in the cited study Medgyesi et al (reference 15), but the authors may consider explicitly summarizing IRB approval and consent in this article as well.

**Results**

-Does the analysis presented match the analysis plan?

-Are the results clearly and completely presented?

-Are the figures (Tables, Images) of sufficient quality for clarity?

Reviewer #1: Statistical Analysis: The authors apply a Multivariate Hurdle model which is briefly discussed in the Appendix. While I was able to understand the mathematical formulation of the model – it would benefit from some typesetting and formatting to better understand the notations. My biggest concern is here is that the authors do not present any statistical results related to the multivariate probit model fit and model adequacy tests – such as the coefficients, standard error, z score or p-values. Although the hurdle model is popular for zero-inflated data, it has several limitations: 1) the two components of the model are assumed to be independent; 2) it does not explicitly capture the relationships between multiple response variables. Given that these two limitations are applicable in this situation – a robustness analysis should be performed. Similarly, the parameters for Zero-inflated Poisson models are not mentioned in Table 1. Without addressing the above limitations, the confidence in the results will be limited.

Some minor points:

“For each behavior and age group we performed a visual observation of the Q-Q plots and selected the simplest distribution that gave a satisfactory fit of the data” – Why was a statistical fit not performed? Please share the Q-Q plot.

Is it possible to quantify the uncertainty in the results due to modelling only five pathogens?

It is reasonable that there is feedback between environmental conditions and behaviour? Therefore, it pi_2 should be conditioned on environmental conditions?

Line 311 – “Data were summarized as frequencies of observed behaviors according to age groups of children per unit of time, rather than frequencies of behaviors for individual child subjects.” – This would lead high sensitivity to outliers – for example one child/toddler or baby can skew the frequency for their age group. To understand this some bootstrapping should be performed.

Reviewer #2: General: throughout the results section data need to be included in the text; as it currently reads more general terms such as "higher" or "small percentage" but these need to be supported by the actual data. The limits of detection of each assay needs to be presented.

Reviewer #3: -Does the analysis presented match the analysis plan?

 The analysis presented matches the analysis plan in the Methods section.

-Are the results clearly and completely presented?

 The results are clearly and completely presented through both written descriptions and visualized figures. See minor comment regarding Figure 2 below.

-Are the figures (Tables, Images) of sufficient quality for clarity?

Figures are of sufficient quality for clarity except Figure 2. The caption of Figure 2, as it is currently written, lists three variables (qPCR gene copies/g, E. coli cfu/g, and number of pathogen gene targets for a given sample), so it is slightly unclear as to which two variables are represented in the correlation coefficients. It appears that the coefficients in the matrix reflect the correlation between either a gene target from qPCR or E. coli cfu/g with other gene target or E. coli cfu/g. It may be beneficial to adjust the caption to clarify exactly which two data sets are being correlated.

**Conclusions**

-Are the conclusions supported by the data presented?

-Are the limitations of analysis clearly described?

-Do the authors discuss how these data can be helpful to advance our understanding of the topic under study?

-Is public health relevance addressed?

Reviewer #1: The results of the models as mentioned above should be made more accessible to the readers. While the authors have listed some key limitations – it also means that these limitations reduce the confidence in results. Effort should be made to quantify the uncertainty due to these limitations.

Reviewer #2: The application of this model and the public health relevance are clear from this manuscript. Several limitations are raised (handling sparse positive data, not carrying the results out to risk/health outcomes, soil samples and observation data not collected at the same time etc.). Perhaps the reasons for not conducting a QMRA or description of the future directions can be elaborated.

Reviewer #3: Are the conclusions supported by the data presented?

Conclusions are supported by the data presented, though “relatively high likelihood of pathogen exposure for children playing in this peri-urban settlement” (Lines 392-393) would be more informative if put into context. Is it high relative to how much pathogen exposure was expected in this sample or high relative to estimated exposure elsewhere?

-Are the limitations of analysis clearly described?

 Limitations are clearly described. Each limitation’s possible impact on the results is explored. The main limitations appear to lead to more conservative estimates of pathogen exposure rather than over-extrapolation of the data.

-Do the authors discuss how these data can be helpful to advance our understanding of the topic under study?

The authors discuss how these data can be helpful to advance the understanding of how children interact with their environment, as well as its implications for pathogen exposure and control in public spaces. The study contributes to the efforts to elucidate the mixed efficacy of household versus public infrastructure interventions. The results also guide future studies toward exploring reasons for the changes in pathogen exposure during development such as behavioral or anatomical changes, implying that WASH interventions may also be customized to the age group of its recipients. The presented model may also be further developed to apply to other types of studies that aim to quantify pathogen exposure as a function of multiple factors.

-Is public health relevance addressed?

Public health relevance is addressed. This study is useful for future studies in modeling pathogen exposure in children playing in public spaces, which can be quantified and adjusted for in future WASH studies.

**Editorial and Data Presentation Modifications?**

Reviewer #1: (No Response)

Reviewer #2: (No Response)

Reviewer #3: Use this section for editorial suggestions as well as relatively minor modifications of existing data that would enhance clarity. If the only modifications needed are minor and/or editorial, you may wish to recommend “Minor Revision” or “Accept”. 

 I recommend “Minor Revision”. For Figure 2 caption: Suggest clearly listing two variables (qPCR gene copies/g or E. coli cfu/g versus number of pathogen gene targets present for a given sample by qPCR) to clarify which two variables are being assessed for correlations.

**Summary and General Comments**

Reviewer #1: To summarize, the goal of this study seems modest related to the current state of art. The authors should focus on addressing some of the limitations they have listed and mentioned here. These limitations are significant and has high sensitivity to the results, The results of the statistical model e.g. hurdle model and ZIP parameters should be made available. Further, a brief overview of the ethics reporting should be provided in the Methods

Reviewer #2: This manuscript presents interesting and timely work investigating multi-pathogen exposures in public play spaces for children. Pairing environmental soil data with context-specific behavioral observations, the authors use a Bayesian normal-hurdle model to estimate the likelihood of single and multi pathogen exposures in one hour. While this work addresses an evidence gap for public play spaces and new modeling approaches for mutiple pathogens at once, additional clarity is needed on the model parameters, reworking the results section to present the numeric results in text, and tightening the discussion section to improve flow.

Introduction:

Line 89: include reference citations earlier on in this sentence rather than at line 92

Lines 89-92: long sentence, recommend breaking into two for improved readability. Also suggest including the specific citations within the sentence when referencing the primary findings. 

Line 92: need to provide a better linkage from the research completed to the need for research into public spaces. I.e. few studies have looked at public spaces, provide the evidence…

Line 108: this paragraph speaks to the time children spend in public spaces but the supporting research presented is frequency in location. Please rephrase 

Methods:

Line 268: “For observation periods of less than one hour, behavior frequencies were extrapolated out to one hour of observation time.” It appears that all? or most of the structured observations were conducted for 30 minutes. Can you expand on this extrapolation, the potential bias this introduces, and why not restrict your analyses to the observed 30 minutes for which you have data? 

Line 285: This is confusing, can you please clarify the number of touches simulated per hour for each of the contacts, e.g. hand to mouth, object to mouth etc., Similarly the number of touches to soil, surfaces

General: were negative controls run with DelAgua? If so, specify frequency of running these and the results (all negative etc., like is done for line 317.

Results

Line 317-318: include the detection frequencies (range) for pathogenic E coli relative to Aeromonas or V. Cholera in the sentence rather than just saying it is higher. 

Table 2: are these detection rates? Looks more like detection frequencies or prevalence: e.g. Aeromonas: 5/79 detected. Suggest specifying more clearly what is being reported under presence. This would need to be updated in the soil contamination section.

Lines 323-324: please expand this sentence, you are indicating that the qPCR data has a lower limit of detection relative to the cultured E coli. Please provide the limits of detection for each assay. I also suggest expanding this section to walk the reader through Figure 1.

Lines 328 - 334: need to include the numerical data to support these phrases; ie. While a small percentage…, what is the percentage? This should be moved to the discussion: “...though this could be due to a combination of organisms/species contributing to the colony count and only one species contributing to the gene copy concentration ”

Lines 355: “Despite having lower presence likelihoods, Aeromonas and EPEC bfpA had the highest average exposure doses across groups.” I am not sure I understand this, if the empirical prevalence for these organisms was the lowest, and you are accounting for the age-stratified behavioral patterns, how is the exposure dose the highest?

Discussion:

Lines 450-453: “The volume of soil used in culture was also four times higher for culture methods compared to qPCR methods, resulting in some discrepancy between the two values and the potential for low concentrations of pathogens to not be detected by qPCR methods”. So this is a point well taken but that is why the authors standardized copies or gene copies/g and as noted earlier the detection limits of the qPCR is more sensitive than the culture. Recommend removing this sentence as it does not advance the discussion section.

Lines 462-464: it was not evident in the methods section that the behavioral and environmental samples were not collected at the same time. While having context specific behavior data is critical for improving exposure estimates, this limitation needs to be caveated in the introduction (e.g. these are not matched soil and observation data in realtime). 

Lines 470: care needs to be taken when describing why QMRA analysis was not conducted here. 1) there needs to be a dose response available for the target pathogens, 2) how the DR are constructed is important as the authors note, and 3) whether the DR include immunity/protective dynamics is a factor. Because the authors are speaking in generalities it is not clear if QMRA would have been feasible for the above 3 reasons. 

Lines 476: “… high contamination environments, the human immune system may respond to increased exposure to pathogens by developing higher resistance, meaning they would need a higher dose of a given pathogen to result in disease or may not develop disease at all.” Need to cite this.

Reviewer #3: The authors present a model to quantify the probability of enteric pathogen exposure from playing in public spaces in children in a peri-urban settlement in Haiti. This study is significant as pathogen exposure in public is often a variable that is not well-characterized yet is likely impactful during WASH intervention delivery and evaluation. Claims are properly placed in the context of previous literature. The authors outline the limited success of prior WASH trials that focused only on the household, evidence of children’s likely exposure to public spaces, and improvements in health outcomes from trials that delivered interventions for public infrastructure. This study is novel as it quantifies risk of pathogen exposure from public spaces as a function of environmental availability of pathogens, child developmental stage, and microbe transfer rate. The authors present a preliminary model that may be further developed and applied to many other scenarios and data sets. Strengths of the study include: combining microbiological, molecular, and behavioral data to quantify risk of pathogen exposure in a low-resource setting, flexibility of the presented model to apply to different settings and data sets; clear writing; and transparent discussion of limitations. However, its main weakness is the lack of mitigating practices for some limitations. For example, the overestimation of viable pathogenic bacteria (Lines 446-450) may be mitigated by checking for RNA expression of the gene targets in bacterial culture, if possible. If not, the authors may briefly describe how future studies may mitigate such limitations. In Lines 397-399, are there citations from other studies demonstrating the plausibility of the reasons that could explain differences by age? This could strengthen the authors’ points. In lines 476-478, please include citations that would support this statement and expand on this concept further. In endemic areas, in addition to resistance, the immune system may have a more robust immunoprotective or immunoregulatory response versus an immunopathologic response. Or in high contamination environments, chronic infection may blunt the immune response. More nuance is needed here.

PLOS authors have the option to publish the peer review history of their article (what does this mean?). If published, this will include your full peer review and any attached files.

Reviewer #1: No

Reviewer #2: No

Reviewer #3: No
---

## [Editor Report · Decision Letter 1]

20 Sep 2024

Dear Dr. Houser,

We are pleased to inform you that your manuscript 'A Multi-Pathogen Behavioral Exposure Model for Young Children Playing in Public Spaces in Developing Communities' has been provisionally accepted for publication in PLOS Neglected Tropical Diseases.

Best regards,

Josh M Colston, Ph.D.

Academic Editor

Justin Remais

Section Editor

Thank you for addressing the comments from the reviewers thoroughly, and congratulations on a well-executed and innovative study that will be of considerable public health importance and interest to the readership of PLOS NTDs.

---

## [Editor Report · Acceptance letter]

2 Oct 2024

Dear Dr. Houser,

We are delighted to inform you that your manuscript, "A Multi-Pathogen Behavioral Exposure Model for Young Children Playing in Public Spaces in Developing Communities," has been formally accepted for publication in PLOS Neglected Tropical Diseases.

Best regards,

Shaden Kamhawi

co-Editor-in-Chief

Paul Brindley

co-Editor-in-Chief
